# Physical Activity of Young Patients following Minimally Invasive Lateral Unicompartmental Knee Replacement

**DOI:** 10.3390/jcm12020635

**Published:** 2023-01-12

**Authors:** Mustafa Hariri, Merlin Hagemann, Paul Mick, Julian Deisenhofer, Benjamin Panzram, Moritz Innmann, Tobias Reiner, Tobias Renkawitz, Tilman Walker

**Affiliations:** Department of Orthopaedic Surgery, University Hospital of Heidelberg, Schlierbacher Landstrasse 200a, 69118 Heidelberg, Germany

**Keywords:** lateral unicompartmental knee arthroplasty, UKR, partial knee replacement, fixed bearing, activity

## Abstract

Unicompartmental knee replacement (UKR) has increased in popularity in recent years, especially in young patients with high demands on their athletic ability. To date, there are no data available on the physical activity of young patients following lateral UKR. The aim of this study was to demonstrate return-to-activity rate and sporting activity of patients aged 60 years or younger following lateral UKR with a fixed-bearing (FB) prosthesis. Thirty-seven patients aged 60 years or younger after lateral FB-UKR were included. Sporting activities were assessed using the University of California Los Angeles activity scale (UCLA) and the Tegner activity score (TAS). Clinical outcome was measured using the Oxford Knee Score (OKS), range of motion (ROM) and visual analogue scale (VAS). The mean follow-up (FU) was 3.1 ± 1.5 years and the mean age at surgery was 52.8 ± 3.1 years. The return-to-activity rate was 87.5% and 49% of patients were highly active postoperatively as defined by an UCLA score of 7 or higher. All clinical parameters increased significantly postoperatively. We demonstrated a high return-to-activity rate with nearly half of the patients achieving high activity levels. Longer FU periods are necessary to evaluate the effect of activity on implant survival.

## 1. Introduction

Unicompartmental knee replacement (UKR) is an effective surgical treatment for end-stage osteoarthritis (OA) limited to a single knee compartment after all joint-preserving methods have failed [1,2]. UKR offers several advantages over total knee replacement (TKR), such as faster recovery, shorter hospital stay and greater range of motion (ROM) [3,4]. Consequently, UKR has gained popularity in the last decade, especially in young and active patients [5,6]. These patients often have high expectations for their postoperative activity level [7]. As unrealistic expectations can lead to disappointments and dissatisfaction, it is important to advise these patients preoperatively based on current evidence [8]. While there are some studies reporting on physical activity after medial UKR, there is a lack of evidence regarding activity levels following lateral UKR due to its relatively rare indication [9,10,11,12]. Because of the anatomic and kinematic differences between the lateral and medial compartments of the knee [13,14,15], data on activity levels of patients following medial UKR are not simply applicable to patients undergoing lateral UKR. Biomechanical analysis using baropodometry in patients after TKR has shown promise in objectifying functional and postural rehabilitation and could also be used in patients undergoing UKR, especially to assess differences between lateral and medial UKR [16].

To date, there are only two studies reporting on physical activity following lateral UKR in general, but there are no data focusing on the postoperative activity of young patients [17,18].

Therefore, the purpose of the present study was to demonstrate the return-to-activity rate and the physical activity of young patients defined as 60 years or younger at the time of surgery following lateral UKR with use of a fixed-bearing (FB) device. We hypothesized that these patients could achieve a high level of activity combined with a good to excellent clinical function.

## 2. Patients and Methods

The current study is based on the retrospective analysis of prospectively collected data from a series of patients who underwent lateral UKR for isolated OA of the knee. A total of 143 UKR were performed in 138 patients between 2013 and 2020 using the Oxford Fixed Lateral prosthesis (Zimmer Biomet UK, Bridgend) at the Department of Orthopaedic Surgery of the University Hospital of Heidelberg. Patients aged 60 years or younger at the time of surgery were enrolled in this study with a minimum follow-up (FU) of 12 months.

Ethical approval was obtained by the institutional review boards of the University of Heidelberg (S-293-2021) and the study was conducted in accordance with the Helsinki Declaration of 1975, as revised in 2013. Informed consent was obtained from all participating patients.

Primary indication was severe OA of the lateral compartment with full-thickness articular cartilage loss (“bone on bone”) or avascular necrosis of the femoral condyle. In all cases, the anterior cruciate ligament (ACL) as well as the MCL and LCL were functionally intact, the valgus deformity was manually correctable and there was no evidence of OA in the medial compartment on varus stress radiographs. OA of the patellofemoral joint was not considered a contraindication unless there was deep eburnation or bone grooving on the medial facet of the patella. Rheumatoid arthritis, fixed valgus deformity, previous osteotomy, or a flexion deformity >15° were considered contraindications [19].

All surgeries were performed using the minimally invasive surgical technique (MIS) through a lateral parapatellar approach without dislocation of the patella. Internal rotation of the tibial plateau and anatomical positioning of the femoral component were respected to avoid an elevation of the joint line. Bearing thickness was selected in full extension, as described previously [19].

Depending on the bone quality, cemented or uncemented fixation was chosen for the femoral component, whereas the tibial component was only available in a cemented version. An intravenous single-shot antibiotic (1.5 g cefuroxime) was administered perioperatively.

Postoperative rehabilitation is standardized for all patients. From the first postoperative day, immediate full weight bearing is possible. No restriction in active and passive knee movement was set. Discharge is followed by 3 weeks of inpatient or outpatient rehabilitation. Regarding postoperative activities, patients were not restricted in their sports engagement but low-impact activities such as biking or Nordic walking were recommended.

All procedures were conducted by or conducted under the supervision of six senior surgeons with extensive experience in knee replacement [19].

Clinical data were obtained through clinical examination by two of the authors (MH, MH) as part of a regular check-up to determine the Oxford knee score (OKS) and range of motion (ROM). These regular check-ups are routinely performed at 1, 3 and 5 years postoperatively in all patients receiving an arthroplasty at our institution. Pain level was assessed using a visual analogue scale (VAS) ranging from 0 to 10 (0 = no pain, 10 = worst pain). Satisfaction was evaluated postoperatively by a numeric scale ranging from 1 (highly satisfied) to 5 (unsatisfied). The University of California Los Angeles activity scale (UCLA), the Tegner activity score (TAS) and the Schulthess Clinic Activity Score were used to assess patients’ physical activity [20,21]. The state of general health was assessed using the Short Form 12 Health Survey (SF-12) [19,22].

Standardized postoperative radiographs were aligned with fluoroscopic control to obtain views parallel to the tibial component in the lateral view. The radiographs were performed immediately postoperatively and at last FU. They were analyzed with a focus on radiological signs of loosening of the components and progression of arthritis in the medial or retropatellar compartment.

### Statistics

Data were recorded and analyzed using SPSS version 20.0 (SPSS Inc., Chicago, IL, USA). The empirical distribution of continuous variables was described using mean and standard deviation. Differences between preoperative and postoperative data were examined with a one-sample *t*-test and Wilcoxon signed-rank test. In case of categorical variables, count and percentage were calculated, and Pearson’s chi-squared test was used to detect differences. For all tests, the significance level was set at *p* < 0.05

## 3. Results

### 3.1. Demographics

A total of 37 patients met the inclusion criteria and were enrolled in this study (Figure 1). One patient was lost to follow-up and no revision surgery was required. The mean FU was 3.1 ± 1.5 years and the mean age at surgery was 52.8 ± 3.1 years. Patient demographics are shown in Table 1.

### 3.2. Reoperation

No revision surgery was performed in our study group, defined as exchange or removal of at least one of the components. Three patients needed additional surgery. One patient suffered a quadriceps tendon rupture 70 days after surgery during rehabilitation and received a surgical repair. One patient had an aseptic wound healing disorder three weeks after surgery and received multiple lavages and debridements without any microbiological or histopathological sign of an infection. One patient demonstrated an early-onset infection with evidence of *E. coli* in the microbiological samples 6 weeks after primary surgery. After the DAIR (debridement and implant retention) procedure, the implant could be retained.

### 3.3. Patient-Reported Outcome Measures

All functional outcome parameters improved significantly from pre- to postoperation.

The mean ROM increased from 121° ± 14.8 to 136° ± 9.3 (*p* < 0.001).

The mean OKS showed an improvement from 26.9 ± 9.0 to 39.9 ± 8.1 (*p* < 0.001). According to the OKS criteria, 59.5% had an excellent outcome (score > 41), 21.6% had a good outcome (34 to 41), 10.8% had a fair outcome (27 to 33) and 8.1% had a poor outcome (<27) at final FU.

The mean pain level on VAS improved from 7 ± 2.1 to 1.7 ± 2.1 (*p* < 0.001).

The UCLA score increased significantly from 3.6 ± 1.8 to 6 ± 1.5 (*p* < 0.001), as did the TAS from 2.2 ± 1.3 to 3.3 ± 1.3 (*p* < 0.001). Forty-nine percent of patients were highly active postoperatively, defined as an UCLA score of 7 or higher.

Overall, 35 patients (94.6%) were satisfied, very satisfied or highly satisfied with the outcome, one patient (2.7%) was fairly satisfied and one patient (2.7%) was unsatisfied (Figure 2). No difference in satisfaction level between females and males was observed (*p* = 0.342).

The mean postoperative SF-12 physical score (PCS) was 43.43 ± 10.7 and the SF-12 mental score (MCS) was 45.97 ± 11.1. Clinical results are demonstrated in Table 2 and Figure 3.

### 3.4. Radiological Outcome

In none of the cases were signs of loosening or progression of OA observed on the most recent radiographs.

### 3.5. Sporting Activities and Participation

Before the onset of the first restricting symptoms, 32 patients (86.5%) were active in at least one physical activity, compared to 30 patients (81.1%) after surgery. Four patients (10.8%) quit their activities entirely after surgery, while two patients (5.4%) started regular physical activities. After surgery, 28 of 32 patients returned to regular physical activity, resulting in a return-to-activity rate of 87.5%. Three (8.1%) of the five patients who were inactive before surgery remained inactive after surgery.

The reasons given for the decrease in their sports ability were “other health issues than knee-related pain” (two patients/50%), “less motivation” (one patient/25%) and “knee pain” (one patient/25%).

Regarding individual physical activities before and after surgery, there was a significant decrease in soccer (*p* = 0.047), while all other activities showed no significant change from pre- to postoperation as shown in Table 3.

### 3.6. Extent of Activities

The extent of activity was assessed by the parameters “number of patients who were active for at least one hour per session” and “number of patients who were active at least three times a week” (Figure 4 and Figure 5). There were no significant changes from pre- to postoperation in these parameters.

Before the onset of the first restricting symptoms as well as after implantation of UKR, 14 patients (37.8%) participated at least three times a week in regular physical activities (*p* = 0.818).

Preoperatively, eight patients (21.6%) were active for at least one hour per session compared to nine patients (24.3%) postoperatively (*p* = 0.715).

A subgroup analysis between female and male patients showed no differences in the session duration (*p* = 0.068), frequency (*p* = 0.781) or number of different sports (*p* = 0.460) postoperation.

Thirty-three patients (89.2%) reported an improvement in sporting activity due to the lateral UKR, while two patients (5.4%) reported no change, and two patients (5.4%) described a worsening of activity level. A total of 15 patients (40.5%) were able to return to regular physical activity within three months after surgery and 21 patients (56.8%) within six months, while 9 patients (24.3%) required more than six months after surgery.

## 4. Discussion

The main finding of this study was that the vast majority of young patients were able to return to their sporting behavior after lateral FB-UKR to which they were accustomed before the onset of restricting symptoms. The return-to-activity rate was 87.5% after a mean FU of 3 years. As stated before, there is a lack of evidence concerning sporting activity after UKR in general and this is the first study reporting on the return to activity in young patients following lateral FB-UKR. Thus, these results must be compared to studies describing activity level for medial UKR and lateral MB-UKR.

To date, there is only one other study reporting on sporting activity following lateral FB-UKR. Zimmerer er al. demonstrated a return-to-activity rate of 86% in 19 patients with a mean FU of 4.6 years [18]. The mean age at the time of surgery was 56.7 years, including patients up to 76 years [18]. While these results are in line with ours, another study from our research group reported on a higher return-to-activity rate of 98% after lateral MB-UKR in a cohort of 45 patients with a mean age of 60.1 years [17]. Comparable results have been reported for medial UKR by Ho et al. with a return-to-activity rate of 87% in 36 patients after 4 years and by Lo Presti et al. with a rate of 90% in 58 patients after 2 years [9,23]. Panzram et al. demonstrated a higher return-to-activity rate of 92.9% in 228 instances of cementless medial MB-UKR after 37.1 months [12], while Pietschmann et al. described a lower rate of 80.1% after 4.2 years in 78 patients receiving a medial UKR [24]. Pietschmann et al. explained the noticeably lower rate in their patients by drawing attention to the higher mean age of 64.4 years at the time of surgery and a natural decline in physical activity in the elderly [24]. Overall, these reported results are higher than those published for patients following TKR ranging from 63.6% to 85% [25,26,27,28], which might be explained due to a better restoration of native limb alignment and physiological knee kinematics after implantation of UKR compared to TKR [3,4,29].

In our cohort, four patients have stopped their activity after UKR. When asked for their reasons, only one patient stated that he had to quit because of knee pain. This patient achieved an OKS of 42 and a ROM of 140 degrees postoperatively. Since no objective reason for his pain such as aseptic loosening, infection, instability or OA progression was identified and the knee function itself was good, we did not perform revision surgery. Unexplained knee pain following knee arthroplasty is the subject of current research and more attention is drawn to component malrotation as a possible explanation. CT scans in patients following TKR showed a correlation between rotational malalignment and unexplained knee pain [30]. Further research on component alignment in patients undergoing UKR is necessary and could help enlighten reasons for unexplained pain.

In the remaining three patients, the reasons for decline in their sport ability were not related to UKR.

Furthermore, there was no significant difference in sport frequency and session length between pre- to postoperation in our study. These results are consistent with previously published data [12,18,25].

Looking at the individual activities, there was a shift from high-impact to low-impact activities postoperatively. A significant change was observed in soccer participation, while there was also a noticeable decrease in other high-impact sports such as tennis and jogging without reaching statistical significance. This decrease in high-impact sports is consistent with the current literature [9,10,18]. Although the reason for this is multifactorial, we assume that the main aspect is surgeons’ recommendations to participate in low-impact sports postoperatively [24,31]. Intense and regular athletic activity may lead to an increase in stress on the implant–bone interface, causing an acceleration in wear and, therefore, can lead to higher revision rates in young and active patients after total joint replacement [17,32]. As the patients who are suitable for implantation of UKR tend to be younger and more active than those suitable for implantation of TKR and the fact that lifetime risk of revision surgery in UKR is higher with younger age at surgery [33], surgeons encourage patients to engage in low-impact activities such as cycling and swimming. However, there is no study reporting on the relationship between use and wear in UKR [10,32,34]. Hence, we think suitable patients who are willing to participate in high-impact activities should not be restricted by the surgeon. In our cohort, three patients were able to regularly participate in high-impact sports such as tennis, soccer or jogging postoperatively and 49% of the patients were highly active, defined by an UCLA score of 7 or higher. Similar proportions of highly active patients following UKR were described earlier [17,35,36]. Further studies with longer FU periods are needed to determine possible effects of activity level on the survivorship of UKR.

Regarding the PROMs in the present study, all parameters improved significantly postoperatively. We demonstrate excellent clinical results with a high mean OKS of 39.9 and 81.1% of our patients having a good to excellent outcome according to the OKS criteria. Comparable results have been reported for patients following lateral FB-UKR [37,38] and lateral MB-UKR [39,40]. The mean postoperative ROM of 136° in our cohort appears to be higher than those of earlier reports of patients following lateral UKR [41,42,43]. Streit et al. demonstrated a mean ROM of 128.6° postoperatively in comparable young patients following medial UKR [44]. Although this ROM is lower than our result, the mean ROM at baseline was also lower than that in our patients: 113.3° compared to 121°. Consequently, the increase in ROM is similar in both studies [44].

The state of general health was assessed using the SF-12 survey which showed results comparable to earlier published data in patients following UKR [18,45].

This study has several limitations. First, we report a small number of patients in a retrospective study design. However, due to the low incidence of isolated lateral knee OA, most studies about lateral UKR have small sample sizes. To date, this study is the first reporting on the activity level of young patients undergoing lateral FB-UKR and it has the largest sample size in terms of physical activity after lateral UKR.

Second, the mean FU of 3 years is too short to analyze possible effects of activity level on prosthesis longevity. Therefore, further studies with longer FU periods are necessary.

In addition, the questionnaire on the physical activities before the onset of the first restricting symptoms was surveyed at the time of the last FU. Therefore, this information dated back several years, causing a hindsight bias.

## 5. Conclusions

This is the first study reporting on activity level in young patients following lateral FB-UKR.

We demonstrate a high return-to-activity rate of 87.5% postoperatively with nearly half of the patients reaching a high activity level defined by an UCLA score of 7 or higher. While most patients participated in low-impact sports postoperatively, there were also some patients who regularly performed in high-impact sports. Furthermore, patients’ satisfaction was very high and all clinical parameters improved significantly. This study provides important information for surgeons and patients regarding postoperative activity level after lateral UKR, especially in young patients.

## Figures and Tables

**Figure 1 jcm-12-00635-f001:**
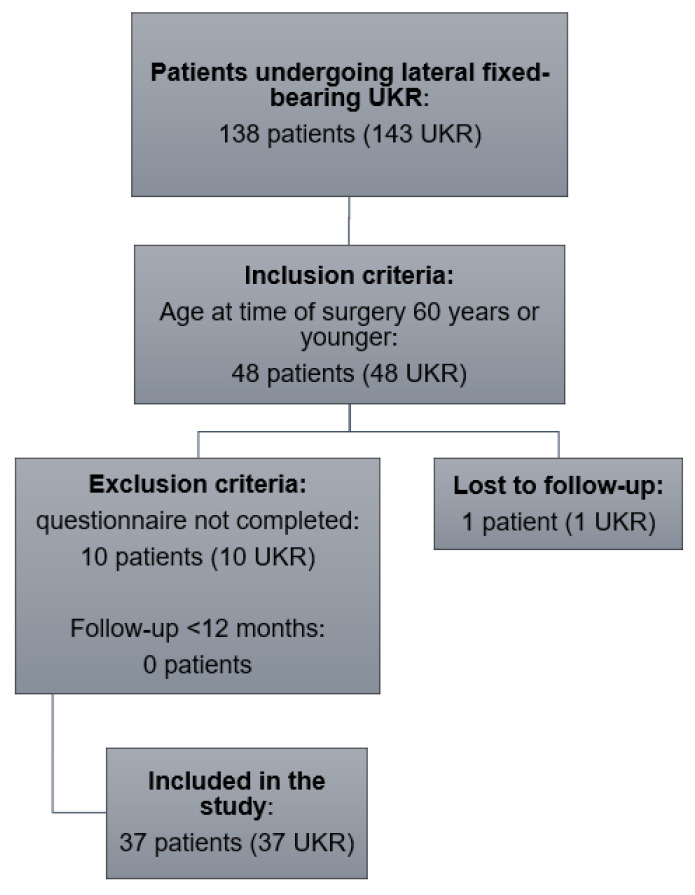
Flowchart illustrating number of patients who met the inclusion and exclusion criteria. UKR: unicompartmental knee replacement.

**Figure 2 jcm-12-00635-f002:**
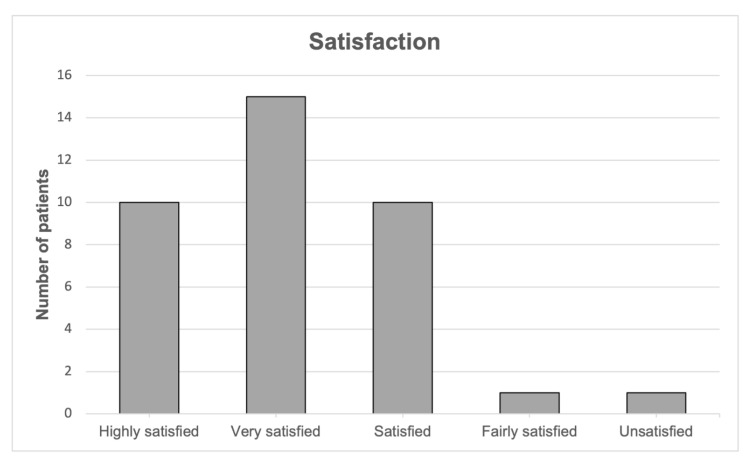
Patient satisfaction at last follow-up.

**Figure 3 jcm-12-00635-f003:**
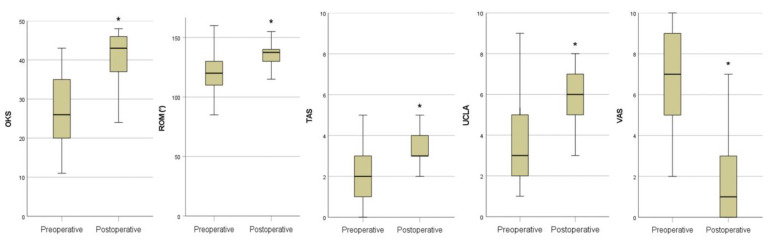
PROMs preoperatively and at last follow-up. * *p* < 0.001. OKS: Oxford knee score; ROM: range of motion; TAS: Tegner activity score; UCLA: University of California Los Angeles activity scale; VAS: visual analogue scale.

**Figure 4 jcm-12-00635-f004:**
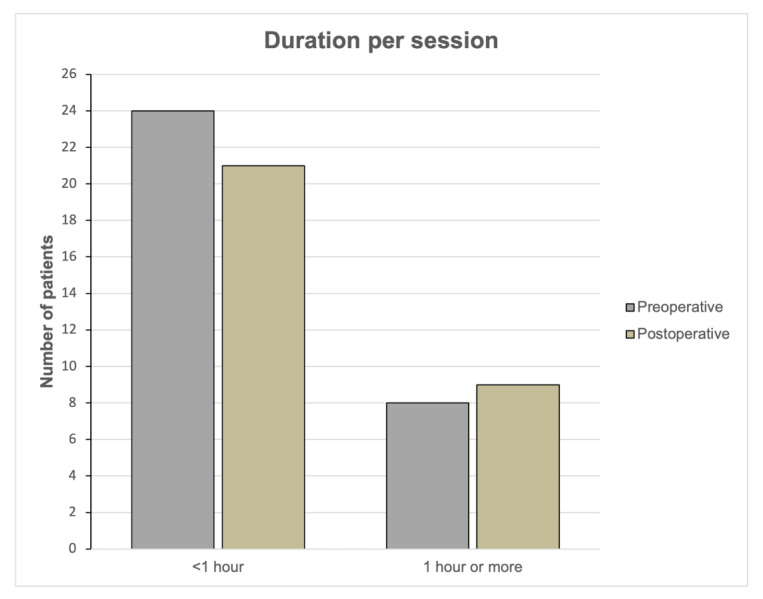
Duration per sport session preoperatively and postoperatively. Bar charts showing the number of patients who were participating less than 1 h or at least 1 h.

**Figure 5 jcm-12-00635-f005:**
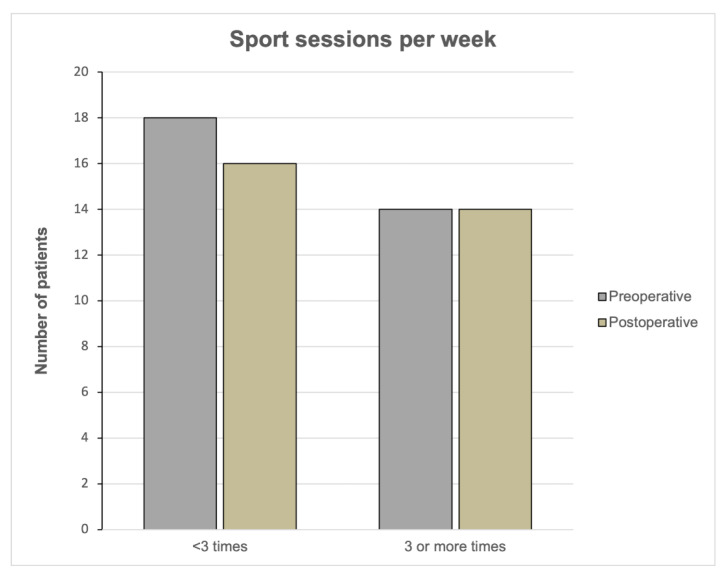
Sport frequency per week preoperatively and postoperatively. Bar charts showing the number of patients who were participating less than 3 times or at least 3 times a week.

**Table 1 jcm-12-00635-t001:** Demographic data of the study group.

Demographics	
Number of patients	37
Mean follow-up in years (±SD)	3.1 ± 1.5
Mean age at time of surgery in years (±SD)	52.8 ± 3.1
Gender (%)	Female 22 (59.5%); Male 15 (40.5%)
Operated side (%)	Left 12 (32.4%); Right 25 (67.6%)
Mean body mass index (kg/m^2^) (±SD)	30.0 ± 6.8

SD: standard deviation.

**Table 2 jcm-12-00635-t002:** PROMs preoperatively and at last follow-up.

	Preoperative (Mean ± SD)	Postoperative (Mean ± SD)
OKS ***	26.9 ± 9.0	39.9 ± 8.1
ROM ***	121 ± 14.8	136 ± 9.3
VAS ***	7 ± 2.1	1.7 ± 2.1
UCLA ***	3.6 ± 1.8	6 + 1.5
TAS ***	2.2 ± 1.3	3.3 ± 1.3
SF12-PCS	-	43.43 ± 10.7
SF12-MCS	-	45.97 ± 11.1

OKS: Oxford knee score; ROM: range of motion; VAS: visual analogue scale; UCLA: University of California Los Angeles activity scale; TAS: Tegner activity score; SF12: Short Form 12 Health Survey; PCS: SF-12 physical score; MCS: SF-12 mental score; SD: standard deviation; *** *p* < 0.001.

**Table 3 jcm-12-00635-t003:** Patients’ physical activities before and after surgery.

Physical Activity	Number of Patients Participating before Surgery (*n*/%)	Number of Patients Participating after Surgery (*n*/%)	Difference (*n*/%)
None	5 (13.5)	7 (18.9)	+2 (+ 5.4)
Biking	23 (62.2)	21 (56.8)	−2 (−5.4)
Hiking	16 (43.2)	13 (35.1)	−3 (−8.1)
Alpine skiing	1 (2.7)	0 (0)	−1 (−2.7)
Jogging	4 (10.8)	1 (2.7)	−3 (−8.1)
Golf	1 (2.7)	1 (2.7)	0 (0)
Soccer	6 (16.2)	1 (2.7)	−5 (−13.5) *
Tennis	2 (5.4)	0 (0)	−2 (−5.4)
Nordic walking	8 (21.6)	8 (21.8)	0 (0)
Hand-, Volley- and Basketball	4 (10.8)	1 (2.7)	−3 (−8.1)
Aqua aerobics	3 (8.1)	1 (2.7)	−2 (−5.4)
Long walks	12 (32.4)	11 (29.7)	−1 (−2.7)
Fitness training	8 (21.6)	6 (16.2)	−2 (−5.4)

* *p* < 0.05.

## Data Availability

The datasets used and analyzed during the current study are available from the corresponding author upon reasonable request.

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
