# Peer review of "Physical Activity of Young Patients following Minimally Invasive Lateral Unicompartmental Knee Replacement"

_jcm, 2023, doi:10.3390/jcm12020635_

Round 1

Reviewer 1 Report

Dear Authors, the topic is estremely interesting.  

As regards the introduction i suggest to improve this section citing a following article at the end of introduction.

- Baropodometry on patients after total knee arthroplasty

Notarnicola A et al.

Musculoskeletal SurgeryVolume 102, Issue 2, Pages 129 - 1371 August 2018

In fact it is important to underline how the instrumental analysis (baropodometry) allows to perform a biomechanical study of active station as reported by Authors. This is important for readers in order to know that according the literature there are a studies in this field

As regards the M&M, i suggest to improve this section by reporting that the Authors collected the x-rays data at different follow-ups in order to check the results and complications such as mobilizations.

The results and discussion are well described and balanced but i suggest to add in the discussion if the Authors find a gender differences in terms of sports activity restart and satisfaction rate .

Furthemore the Authors have to discuss the relationship between painfl knee prosthesis and satisfaction rate citing also the following article:

-Painful knee prosthesis: CT scan to assess patellar angle and implant malrotation

Spinarelli A. et al.

Muscles, Ligaments and Tendons JournalOpen AccessVolume 6, Issue 4, Pages 461 - 4661 October 2016

As regards the conclusion it is clear  

Author Response

Point 1: As regards the introduction i suggest to improve this section citing a following article at the end of introduction.

 - Baropodometry on patients after total knee arthroplasty

Notarnicola A et al.

Musculoskeletal SurgeryVolume 102, Issue 2, Pages 129 - 1371 August 2018

 In fact it is important to underline how the instrumental analysis (baropodometry) allows to perform a biomechanical study of active station as reported by Authors. This is important for readers in order to know that according the literature there are a studies in this field

Response 1: This is an interesting study reporting about a promising instrumental support. We included this citation in our introduction as suggested (Line 47-50).

Point 2: As regards the M&M, i suggest to improve this section by reporting that the Authors collected the x-rays data at different follow-ups in order to check the results and complications such as mobilizations.

Response 2: Thank you for this remark, we have added this information in the materials (Line 113-117) and results (Line 176-177).

Point 3: The results and discussion are well described and balanced but i suggest to add in the discussion if the Authors find a gender differences in terms of sports activity restart and satisfaction rate .

Response 3: We performed a subgroup analysis between females and males and found no differences in terms of satisfaction level or extent of activities postoperative. We have added the results in Line 158-159 and Line 204-206.

Point 4: Furthemore the Authors have to discuss the relationship between painfl knee prosthesis and satisfaction rate citing also the following article:

 -Painful knee prosthesis: CT scan to assess patellar angle and implant malrotation

Spinarelli A. et al.

Muscles, Ligaments and Tendons JournalOpen AccessVolume 6, Issue 4, Pages 461 - 4661 October 2016

Response 4: Thank you for this suggestion. We have added this citation in the discussion section (Line 249-255). Since total knee replacement was investigated in this study, we consider the results to be transferable only to a limited extent. Nevertheless they offer interesting insides into future research topics.  

Reviewer 2 Report

The authors investigated sports activities after lateral type UKA. They have already revealed the sports activity after 50 consecutive lateral UKAs in 2015. However, I examined the duration of surgery in the cases reviewed, maybe no cases overlapped in both manuscripts. Since there are few papers evaluated sports activity after lateral UKA, I recognized the results presented in presented manuscript would be of value to JCM readers. However, minor revision would be required before publication.

My suggestions are as follows:

 Line 47: References 13 and 14 are manuscripts revealed knee kinematics, therefor they are not appropriate to cite in this sentence.

 Line 60: The Oxford Fixed Lateral Prosthesis is a product of Zimmer Biomet.

 Postoperative patient activity is strongly influenced by the physician's recommendations. In patients and methods, the authors should describe postoperative rehabilitation and the physician's recommendations for sports activities after lateral UKA.

 Are there any cases that required revision surgery, additional surgery, or showed component loosening on imaging during the observation period?

Author Response

Point 1: Line 47: References 13 and 14 are manuscripts revealed knee kinematics, therefor they are not appropriate to cite in this sentence.

Response 1: Thank you for that comment. We have changed the term from “biomechanics” to “kinematics” and added a reference to a study by Baré et al in which they demonstrate the different anatomy of the lateral convex tibial plateau compared to the concave medial plateau. The study by Tokuhara et al showed higher laxity of the lateral collateral ligaments, resulting in a higher flexion gap of the lateral knee compartment. Nakagawa et al showed a higher backward movement of the lateral femoral condyle in deep flexion than of the medial condyle. We believe both studies contributed important information to our understanding of the biomechanic in the lateral knee compartment and would like to retain these citations in this section.

Point 2: Line 60: The Oxford Fixed Lateral Prosthesis is a product of Zimmer Biomet.

Response 2: You are right. We changed the branding as recommended (Line 63).

Point 3: Postoperative patient activity is strongly influenced by the physician's recommendations. In patients and methods, the authors should describe postoperative rehabilitation and the physician's recommendations for sports activities after lateral UKA.

Response 3: Thank you for this suggestion. We have added the recommended description in this section (Line 88-93).

Point 4: Are there any cases that required revision surgery, additional surgery, or showed component loosening on imaging during the observation period?

Response 4: No revision surgery defined as exchange or removal of at least one component was performed. 3 patients needed additional surgery after which the implant could be retained.

 Radiographic analysis showed no signs of loosening or progression of osteoarthritis. We have added this information in the results (Line 134-143; Line 176-177) and methods section (Line 113-117)

Round 2

Reviewer 1 Report

Dear Authors , your revisions are acceptable and you followed my suggestion